# Batch Multi-Fidelity Bayesian Optimization with Deep Auto-Regressive Networks

**Shibo Li, Robert M. Kirby, and Shandian Zhe**
School of Computing, University of Utah
Salt Lake City, UT 84112
shibo@cs.utah.edu, kirby@cs.utah.edu, zhe@cs.utah.edu

## Abstract

Bayesian optimization (BO) is a powerful approach for optimizing black-box, expensive-to-evaluate functions. To enable a flexible trade-off between the cost and accuracy, many applications allow the function to be evaluated at different fidelities. In order to reduce the optimization cost while maximizing the benefit-cost ratio, in this paper we propose Batch Multi-fidelity Bayesian Optimization with Deep Auto-Regressive Networks (BMBO-DARN). We use a set of Bayesian neural networks to construct a fully auto-regressive model, which is expressive enough to capture strong yet complex relationships across *all* the fidelities, so as to improve the surrogate learning and optimization performance. Furthermore, to enhance the quality and diversity of queries, we develop a simple yet efficient batch querying method, without any combinatorial search over the fidelities. We propose a batch acquisition function based on Max-value Entropy Search (MES) principle, which penalizes highly correlated queries and encourages diversity. We use posterior samples and moment matching to fulfill efficient computation of the acquisition function, and conduct alternating optimization over every fidelity-input pair, which guarantees an improvement at each step. We demonstrate the advantage of our approach on four real-world hyperparameter optimization applications.

## 1 Introduction

Many applications demand we optimize a complex function of an unknown form that is expensive to evaluate. Bayesian optimization (Mockus, 2012; Snoek et al., 2012) is a powerful approach to optimize such functions. The key idea is to use a probabilistic surrogate model, typically Gaussian processes (Rasmussen and Williams, 2006), to iteratively approximate the target function, integrate the posterior information to compute and maximize an acquisition function so as to generate new inputs at which to query, update the model with new examples, and meanwhile approach the optimum.

In practice, to enable a flexible trade-off between the computational cost and accuracy, many applications allow us to evaluate the target function at different fidelities. For example, to evaluate the performance of the hyperparameters for a machine learning model, we can train the model thoroughly, *i.e.*, with sufficient iterations/epochs, to obtain the accurate evaluation (high-fidelity yet often costly) or just run a few iterations/epochs to obtain a rough estimate (low-fidelity but much cheaper).

Many multi-fidelity BO algorithms (Lam et al., 2015; Kandasamy et al., 2016; Zhang et al., 2017; Song et al., 2019; Takeno et al., 2019) have therefore been proposed to identify both the fidelities and inputs at which to query, so as to reduce the cost and achieve a good benefit-cost balance. Notwithstanding their success, these methods often overlook the strong yet complex relationships between different fidelities or adopt an over-simplified assumption, (partly) for the sake of convenience in calculating/maximizing the acquisition function considering fidelities. This, however, can restrict the performance of the surrogate model, impair the optimization efficiency and increase the cost. For

35th Conference on Neural Information Processing Systems (NeurIPS 2021).

example, Lam et al. (2015); Kandasamy et al. (2016) learned an independent GP for each fidelity, Zhang et al. (2017) used multitask GPs with a convolved kernel for multi-fidelity modeling and have to use a simple smoothing kernel (*e.g.*, Gaussian) for tractable convolutions. The recent work (Takeno et al., 2019) imposes a linear correlation across different fidelities. In addition, the standard one-by-one querying strategy needs to sequentially run each query and cannot utilize parallel computing resources to accelerate, *e.g.*, multi-core CPUs/GPUs and clusters. While incrementally absorbing more information, it does not explicitly account for the correlation between different queries, hence still has a risk to bring in highly correlated examples that includes redundant information.

To address these issues, we propose BMBO-DARN, a novel batch multi-fidelity Bayesian optimization method. First, we develop a deep auto-regressive model to integrate training examples at various fidelities. Each fidelity is modeled by a Bayesian neural network (NN), where the output predicts the objective function value at that fidelity and the input consists of the original inputs and the outputs of *all* the previous fidelities. In this way, our model is adequate to capture the complex, strong correlations (*e.g.*, nonstationary, highly nonlinear) across all the fidelities to enhance the surrogate learning. We use Hamiltonian Monte-Carlo (HMC) sampling for posterior inference. Next, to improve the quality of the queries, we develop a simple yet efficient method to jointly fetch a batch of inputs and fidelities. Specifically, we propose a batch acquisition function based on the state-of-the-art Max-value Entropy Search (MES) principle (Wang and Jegelka, 2017). The batch acquisition function explicitly penalizes highly correlated queries and encourages diversity. To efficiently compute the acquisition function, we use the posterior samples of the NN weights and moment matching to construct a multi-variate Gaussian posterior for all the fidelity outputs and the function optimum. To prevent a combinatorial search over multiple fidelities in maximizing the acquisition function, we develop an alternating optimization algorithm to cyclically update each pair of input and fidelity, which is much more efficient and guarantees an improvement at each step.

For evaluation, we examined BMBO-DARN in both synthetic benchmarks and real-world applications. The synthetic benchmark tasks show that given a small number of training examples, our deep auto-regressive model can learn a more accurate surrogate of the target function than other state-of-the-art multi-fidelity BO models. We then evaluated BMBO-DARN on four popular machine learning models (CNN, online LDA, XGBoost and Physics informed NNs) for hyperparameter optimization. BMBO-DARN can find more effective hyperparameters leading to superior predictive performance, and meanwhile spends smaller total evaluation costs, as compared with state-of-the-art multi-fidelity BO algorithms and other popular hyperparameter tuning methods.

## 2 Background

**Bayesian Optimization (BO)** (Mockus et al., 1978; Snoek et al., 2012) is a popular approach for optimizing black-box functions that are often costly to evaluate and cannot provide exact gradient information. BO learns a probabilistic surrogate model to predict the function value across the input space and quantifies the predictive uncertainty. At each step, we use this information to compute an acquisition function to measure the utility of querying at different inputs. By maximizing the acquisition function, we find the next input at which to query, which is supposed to be closer to the optimum. Then we add the new example into the training set to improve the accuracy of the surrogate model. The procedure is repeated until we find the optimal input or the maximum number of queries have been finished. There are a variety of acquisition functions, such as Expected Improvement (EI) (Mockus et al., 1978) and Upper Confidence Bound (UCB) (Srinivas et al., 2010). The recent state-of-the-art addition is Maximum-value Entropy Search (MES) (Wang and Jegelka, 2017),

$$a(\mathbf{x}) = \mathbb{I}\big(f(\mathbf{x}), f^*|\mathcal{D}\big), \tag{1}$$

where $\mathbb{I}(\cdot, \cdot)$ is the mutual information, $f(\mathbf{x})$ is the objective function value at $\mathbf{x}$, $f^*$ the minimum, and $\mathcal{D}$ the training data collected so far for the surrogate model. Note that both $f(\mathbf{x})$ and $f^*$ are considered as generated by the posterior of the surrogate model given $\mathcal{D}$; they are random variables.

The most commonly used class of surrogate models is Gaussian process (GP) (Rasmussen and Williams, 2006). Given the training dataset $\mathbf{X} = [\mathbf{x}_1^\top, \ldots, \mathbf{x}_N^\top]^\top$ and $\mathbf{y} = [y_1, \ldots, y_N]^\top$, a GP assumes the outputs $\mathbf{y}$ follow a multivariate Gaussian distribution, $p(\mathbf{y}|\mathbf{X}) = \mathcal{N}(\mathbf{y}|\mathbf{m}, \mathbf{K} + v\mathbf{I})$, where $\mathbf{m}$ is the mean function values at the inputs $\mathbf{X}$, often set to $\mathbf{0}$, $v$ is the noise variance, and $\mathbf{K}$ is a kernel matrix on $\mathbf{X}$. Each $[\mathbf{K}]_{ij} = \kappa(\mathbf{x}_i, \mathbf{x}_j)$, where $\kappa(\cdot, \cdot)$ is a kernel function. For example, a popular one is the RBF kernel, $\kappa(\mathbf{x}_i, \mathbf{x}_j) = \exp\big(-\beta^{-1}\|\mathbf{x}_i - \mathbf{x}_j\|^2\big)$. An important advantage of GPs

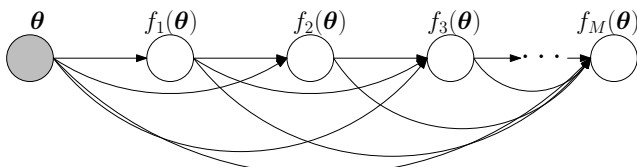

Figure 1: Graphical representation of the deep auto-regressive model in BMBO-DARN. The output at each fidelity $f_m(\mathbf{x})$ ($1 \le m \le M$) is calculated by a (deep) neural network.

is their convenience in uncertainty quantification. Since GPs assume any finite set of function values follow a multi-variate Gaussian distribution, given a test input $\hat{\mathbf{x}}$, we can compute the predictive (or posterior) distribution $p\left(f(\hat{\mathbf{x}})|\hat{\mathbf{x}}, \mathbf{X}, \mathbf{y}\right)$ via a conditional Gaussian distribution, which is simple and analytical.

**Multi-Fidelity BO.** Since evaluating the exact value of the object function is often expensive, many practical applications provide multi-fidelity evaluations $\{f_1(\mathbf{x}), \ldots, f_M(\mathbf{x})\}$ to allow us to choose a trade-off between the accuracy and cost. Accordingly, many multi-fidelity BO algorithms have been developed to select both the inputs and fidelities to reduce the cost and to achieve a good balance between the optimization progress and cost, *i.e.*, the benefit-cost ratio. For instance, MF-GP-UCB (Kandasamy et al., 2016) sequentially queries at each fidelity (from the lowest one, *i.e.*, $m = 1$) until the confidence band is over a given threshold. In spite of its great success and guarantees in theory, MF-GP-UCB uses a set of independent GPs to estimate the objective at each fidelity, and hence ignores the valuable correlations between different fidelities. MF-PES (Zhang et al., 2017) uses a multi-task GP surrogate where each task corresponds to one fidelity, and convolves a smoothing kernel with the kernel of a shared latent function to obtain the cross-covariance. The recent MF-MES (Takeno et al., 2019) also builds a multi-task GP surrogate, where the covariance function is

$$\kappa\left(f_m(\mathbf{x}), f_{m'}(\mathbf{x}')\right) = \sum_{j=1}^{d} \left(u_{mj}u_{m'j} + \mathbb{1}(m = m') \cdot \alpha_{mj}\right) \rho_j(\mathbf{x}_1, \mathbf{x}_2), \qquad (2)$$

where $\alpha_{mj} > 0$, $\mathbb{1}(\cdot)$ is the indicator function, $\{u_{mj}\}_{j=1}^{d}$ is $d$ latent features for each fidelity $m$, and $\{\rho_j(\cdot, \cdot)\}$ are $d$ bases kernels, usually chosen as a commonly used stationary kernel, *e.g.*, RBF.

## 3 Deep Auto-Regressive Model for Multi-Fidelity Surrogate Learning

Notwithstanding the elegance and success of the existing multi-fidelity BO methods, they often ignore or oversimplify the complex, strong correlations between different fidelities, and hence can be inefficient for surrogate learning, which might further lower the optimization efficiency and incur more expenses. For example, the state-of-the-art methods MF-GP-UCB (Kandasamy et al., 2016) estimate a GP surrogate for each fidelity independently; MF-PES (Zhang et al., 2017) has to adopt a simple form for both the smoothing and latent function kernel (*e.g.*, Gaussian and delta) to achieve an analytically tractable convolution, which might limit the expressivity in estimating the cross-fidelity covariance; MF-MES (Takeno et al., 2019) essentially imposes a linear correlation structure between different fidelities — for any input $\mathbf{x}$, $\kappa\left(f_m(\mathbf{x}), f_{m'}(\mathbf{x})\right) = \mathbf{u}_{m_1}^{\top}\mathbf{u}_{m_2} + \alpha_m$ where $\mathbf{u}_m = [u_{m1}, \ldots, u_{md}]$ and $\widetilde{\alpha}_m = \sum_{j=1}^{d} \alpha_{mj}$ if we use a RBF basis kernel (see (2)).

To overcome this limitation, we develop a deep auto-regressive model for multi-fidelity surrogate learning. Our model is expressive enough to capture the strong, possibly very complex (*e.g.*, highly nonlinear, nonstationary) relationships between all the fidelities to improve the prediction (at the highest fidelity). As such, our model can more effectively integrate multi-fidelity training information to better estimate the objective function.

Specifically, given $M$ fidelities, we introduce a chain of $M$ neural networks, each of which models one fidelity and predicts the target function at that fidelity. Denote by $\mathbf{x}_m$, $\mathcal{W}_m$, and $\psi_{\mathcal{W}_m}(\cdot)$ the NN input, parameters and output mapping at each fidelity $m$. Our model is defined as follows,

$$\mathbf{x}_m = [\mathbf{x}; f_1(\mathbf{x}); \ldots; f_{m-1}(\mathbf{x})],$$
$$f_m(\mathbf{x}) = \psi_{\mathcal{W}_m}(\mathbf{x}_m), \quad y_m(\mathbf{x}) = f_m(\mathbf{x}) + \epsilon_m, \qquad (3)$$

where $\mathbf{x}_1 = \mathbf{x}$, $f_m(\mathbf{x})$ is the prediction (*i.e.*, NN output) at the $m$-th fidelity, $y_m(\mathbf{x})$ is the observed function value, and $\epsilon_m$ is a random noise, $\epsilon_m \sim \mathcal{N}(\epsilon_m|0, \tau_m^{-1})$. We can see that each input $\mathbf{x}_m$

consists of not only the original input $\mathbf{x}$ of the objective function, but also the outputs from *all* the previous fidelities. Via a series of linear projection and nonlinear activation from the NN, we obtain the output at fidelity $m$. In this way, our model fully exploits the information from the lower fidelities and can flexibly capture arbitrarily complex relationships between the current and *all* the previous fidelities by learning an NN mapping, $f_m(\mathbf{x}) = \psi_{\mathcal{W}_m}\left(\mathbf{x}_m, f_1(\mathbf{x}), \ldots, f_{m-1}(\mathbf{x})\right)$.

We assign a standard Gaussian prior distribution over each element of the NN parameters $\mathcal{W} = \{\mathcal{W}_1, \ldots, \mathcal{W}_M\}$, and a Gamma prior over each noise precision, $p(\tau_m) = \text{Gam}(\tau_m | a_0, b_0)$. Given the dataset $\mathcal{D} = \{\{(\mathbf{x}_{nm}, y_{nm})\}_{n=1}^{N_m}\}_{m=1}^M$, the joint probability of our model is given by

$$p(\mathcal{W}, \tau, \mathcal{Y}, \mathcal{S}|\mathcal{X}) = \mathcal{N}(\text{vec}(\mathcal{W})|\mathbf{0}, \mathbf{I}) \prod_{m=1}^M \text{Gam}(\tau_m|a_0, b_0) \prod_{m=1}^M \prod_{n=1}^{N_m} \mathcal{N}\left(y_{nm}|f_m(\mathbf{x}_{nm}), \tau_m^{-1}\right), \quad (4)$$

where $\tau = [\tau_1, \ldots, \tau_M]$, $\mathcal{X} = \{\mathbf{x}_{nm}\}$, $\mathcal{Y} = \{y_{nm}\}$, and $\text{vec}(\cdot)$ is vectorization. The graphical representation of our model is given in Fig. 1. We use Hamiltonian Monte Carlo (HMC) (Neal et al., 2011) sampling to perform posterior inference due to its unbiased, high-quality uncertainty quantification, which is critical to calculate the acquisition function. However, our method allows us to readily switch to other approximate inference approaches as needed (see Sec. 4), *e.g.*, stochastic gradient HMC (Chen et al., 2014) used in the excellent work of Springenberg et al. (2016).

## 4 Batch Acquisition for Multi-Fidelity Optimization

Given the posterior of our model, we aim to compute and optimize an acquisition function to identify the input and fidelity at which to query next. Popular BO methods query at one input each time and then update the surrogate model. While successful, this one-by-one strategy has to run each query sequentially and cannot take advantage of parallel computing resources (that are often available in practice) to further accelerate, such as multi-core CPU and GPU workstations and computer clusters. In addition, the one-by-one strategy although gradually integrates more data information, it lacks an explicit mechanism to take into account the correlation across different queries, hence still has a risk to bring in highly correlated examples with redundant information, especially in the multi-fidelity setting, *e.g.*, querying at the same input with another fidelity. To allow parallel query and to improve the query quality and diversity, we develop a batch acquiring approach to jointly identify a set of inputs and fidelities at a time, presented as follows.

### 4.1 Batch Acquisition Function

We first propose a batch acquisition function based on the MES principle (Zhang et al., 2017) (see (1)). Denote by $B$ the batch size and by $\{\lambda_1, \ldots, \lambda_M\}$ the cost of querying at $M$ fidelities. We want to jointly identify $B$ pairs of inputs and fidelities $(\mathbf{x}_1, m_1), \ldots, (\mathbf{x}_B, m_B)$ at which to query. The batch acquisition function is given by

$$a_{\text{batch}}(\mathbf{X}, \mathbf{m}) = \frac{\mathbb{I}(\{f_{m_1}(\mathbf{x}_1), \ldots, f_{m_B}(\mathbf{x}_B)\}, f^*|\mathcal{D})}{\sum_{k=1}^B \lambda_{m_k}}, \quad (5)$$

where $\mathbf{X} = \{\mathbf{x}_1, \ldots, \mathbf{x}_B\}$ and $\mathbf{m} = [m_1, \ldots, m_B]$. As we can see, our batch acquisition function explicitly penalizes highly correlated queries, encouraging joint effectiveness and diversity — if between the outputs $\{f_{m_k}(\mathbf{x}_k)\}_{k=1}^B$ are high correlations, the mutual information in the numerator will decrease. Furthermore, by dividing the total querying cost in (5), the batch acquisition function expresses a balance between the benefit of these queries (in probing the optimum) and the price, *i.e.*, benefit-cost ratio. When we set $B = 1$, our batch acquisition function is reduced to the single one used in (Takeno et al., 2019).

### 4.2 Efficient Computation

Given $\mathbf{X}$ and $\mathbf{m}$, the computation of (5) is challenging, because it involves the mutual information between a set of NN outputs and the function optimum. To address this challenge, we use posterior samples and moment matching to approximate $p(\mathbf{f}, f^*|\mathcal{D})$ as a multi-variate Gaussian distribution, where $\mathbf{f} = [f_{m_1}(\mathbf{x}_1), \ldots, f_{m_B}(\mathbf{x}_B)]$. Specifically, we first draw a posterior sample of the NN weights $\mathcal{W}$ from our model. We then calculate the output at each input and fidelity to obtain a sample of $\mathbf{f}$, and maximize (or minimize) $f_M(\cdot)$ to obtain a sample of $f^*$. We use L-BFGS (Liu and Nocedal,

1989) for optimization. After we collect $L$ independent samples $\{(\widehat{\mathbf{f}}_1, \widehat{f}_1^*), \ldots, (\widehat{\mathbf{f}}_L, \widehat{f}_L^*)\}$, we can estimate the first and second moments of $\mathbf{h} = [\mathbf{f}; f^*]$, namely, mean and covariance matrix,

$$\boldsymbol{\mu} = \frac{1}{L} \sum_{j=1}^{L} \widehat{\mathbf{h}}_j, \quad \boldsymbol{\Sigma} = \frac{1}{L-1} \sum_{j=1}^{L} (\widehat{\mathbf{h}}_j - \boldsymbol{\mu})(\widehat{\mathbf{h}}_j - \boldsymbol{\mu})^\top,$$

where each $\widehat{\mathbf{h}}_j = [\widehat{\mathbf{f}}_j; \widehat{f}_j^*]$. We then use these moments to match a multivariate Gaussian posterior, $p(\mathbf{h}|\mathcal{D}) \approx \mathcal{N}(\mathbf{h}|\boldsymbol{\mu}, \boldsymbol{\Sigma})$. Then the mutual information can be computed with a closed form,

$$\mathbb{I}(\mathbf{f}, f^*|\mathcal{D}) = \mathbb{H}(\mathbf{f}|\mathcal{D}) + \mathbb{H}(f^*|\mathcal{D}) - \mathbb{H}(\mathbf{f}, f^*|\mathcal{D}) \approx \frac{1}{2} \log |\boldsymbol{\Sigma}_{\mathbf{ff}}| + \frac{1}{2} \log \sigma_{**} - \frac{1}{2} \log |\boldsymbol{\Sigma}|, \quad (6)$$

where $\boldsymbol{\Sigma}_{\mathbf{ff}} = \boldsymbol{\Sigma}[1:B, 1:B]$, *i.e.*, the first $B \times B$ sub-matrix along the diagonal, which is the posterior covariance of $\mathbf{f}$, and $\sigma_{**} = \boldsymbol{\Sigma}[B+1, B+1]$, *i.e.*, the posterior variance of $f^*$. The batch acquisition function is therefore calculated from

$$a_{\text{batch}}(\mathbf{X}, \mathbf{m}) \approx \frac{1}{2 \sum_{k=1}^{B} \lambda_{m_k}} \left( \log |\boldsymbol{\Sigma}_{\mathbf{ff}}| + \log \sigma_{**} - \log |\boldsymbol{\Sigma}| \right). \quad (7)$$

Note that $\boldsymbol{\Sigma}$ is a function of the inputs $\mathbf{X}$ and fidelities $\mathbf{m}$ and hence so are its submatrix and elements, $\boldsymbol{\Sigma}_{\mathbf{ff}}$ and $\sigma_{**}$. To obtain a reliable estimate of the moments, we set $L = 100$ in our experiments. Note that our method can be applied along with any posterior inference algorithm, such as variational inference and SGHMC (Chen et al., 2014), as long as we can generate posterior samples of the NN weights, not restricted to the HMC adopted in our paper.

### 4.3 Optimizing a Batch of Fidelities and Inputs

Now, we consider maximizing (7) to identify $B$ inputs $\mathbf{X}$ and their fidelities $\mathbf{m}$ at which to query. However, since the optimization involves a mix of continuous inputs and discrete fidelities, it is quite challenging. A straightforward approach would be to enumerate all possible configurations of $\mathbf{m}$, for each particular configuration, run a gradient based optimization algorithm to find the optimal inputs, and then pick the configuration and its optimal inputs that give the largest value of the acquisition function. However, doing so is essentially conducting a combinatorial search over $B$ fidelities, and the search space grows exponentially with $B$, *i.e.*, $\mathcal{O}(M^B) = \mathcal{O}(e^{B \log M})$. Hence, it will be very costly, even infeasible for a moderate choice of $B$.

To address this issue, we develop an alternating optimization algorithm. Specifically, we first initialize all the $B$ queries, $\mathcal{Q} = \{(\mathbf{x}_1, m_1), \ldots, (\mathbf{x}_B, m_B)\}$, say, randomly. Then each time, we only optimize one pair of the input and fidelity $(\mathbf{x}_k, m_k)(1 \le k \le B)$, while fixing the others. We cyclically update each pair, where each update is much cheaper but guarantees to increase $a_{\text{batch}}$. Specifically, each time, we maximize

$$a_{\text{batch},k}(\mathbf{x}, m) = \frac{\mathbb{I}(\mathcal{F}_{\neg k} \cup \{f_m(\mathbf{x})\}, f^*|\mathcal{D})}{\lambda_m + \sum_{j \neq k} \lambda_{m_j}}, \quad (8)$$

where $\mathcal{F}_{\neg k} = \{f_{m_j}(\mathbf{x}_j) | j \neq k\}$. Note that the computation of (8) still follows (7). We set $(\mathbf{x}_k, m_k)$ to the optimum $(\mathbf{x}^*, m^*)$, and then proceed to optimize the next input location and fidelity $(\mathbf{x}_{k+1}, m_{k+1})$ in $\mathcal{Q}$ with the others fixed. We continues this until we finish updating all the queries in $\mathcal{Q}$, which corresponds to one iteration. We can keep running iterations until the increase of the batch acquisition function is less than a tolerance level or a maximum number of iterations has been done. Suppose we ran $G$ iterations, the time complexity is $\mathcal{O}(GMB)$, which is linear in the number of fidelities and batch size, and hence is much more efficient than the naive combinatorial search. Our multi-fidelity BO approach is summarized in Algorithm 1.

## 5 Related Work

Most Bayesian optimization (BO) (Mockus, 2012; Snoek et al., 2012) methods are based on Gaussian processes (GPs) and a variety of acquisition functions, such as (Mockus et al., 1978; Auer, 2002; Srinivas et al., 2010; Hennig and Schuler, 2012; Hernández-Lobato et al., 2014; Wang and Jegelka, 2017; Kandasamy et al., 2017b; Garrido-Merchán and Hernández-Lobato, 2020). Snoek et al. (2015) showed Bayesian neural networks (NNs) can also be used as a general surrogate model, and has

---

**Algorithm 1** BMBO-DARN ($\mathcal{D}$, $B$, $M$, $T$, $\{\lambda_m\}_{m=1}^M$ )

---

Learn the deep auto-regressive model (4) on $\mathcal{D}$ with HMC.
**for** $t = 1, \ldots, T$ **do**
    Collect a batch of $B$ queries, $\mathcal{Q} = \{(\mathbf{x}_k, m_k)\}_{k=1}^B$, with Algorithm 2.
    Query the objective function value at each input $\mathbf{x}_k$ and fidelity $m_k$ in $\mathcal{Q}$
    $\mathcal{D} \leftarrow \mathcal{D} \cup \{(\mathbf{x}_k, y_k, m_k) | 1 \leq k \leq B\}$.
    Re-train the deep auto-regressive model on $\mathcal{D}$ with HMC.
**end for**

---

**Algorithm 2** BatchAcquisition($\{\lambda_m\}$, $B$, $L$, $G$, $\xi$)

---

Initialize $\mathcal{Q} = \{(\mathbf{x}_1, m_1), \ldots, (\mathbf{x}_B, m_B)\}$ randomly.
Collect $L$ independent posterior samples of the NN weights.
**repeat**
    **for** $k = 1, \ldots, B$ **do**
        Use the posterior samples to calculate and optimize (8),

$$(\mathbf{x}^*, m^*) = \operatorname*{argmax}_{\mathbf{x} \in \Omega, 1 \leq m \leq M} a_{\text{batch},k}(\mathbf{x}, m),$$

        where $\Omega$ is the input domain.
        $(\mathbf{x}_k, m_k) \leftarrow (\mathbf{x}^*, m^*)$.
    **end for**
**until** $G$ iterations are done or the increase of $a_{\text{batch}}$ in (7) is less than $\xi$
Return $\mathcal{Q}$.

---

excellent performance. Moreover, the training of NNs is scalable, not suffering from $\mathcal{O}(N^3)$ time complexity ($N$ is the number of examples) of training exact GPs. Springenberg et al. (2016) further used scale adaption to develop a robust stochastic gradient HMC for the posterior inference in the NN based BO. Recent works that deal with discrete inputs (Baptista and Poloczek, 2018) or mixed discrete and continuous inputs (Daxberger et al., 2019) use an explicit nonlinear feature mapping and Bayesian linear regression, which can be viewed as one-layer Bayesian NNs.

There have been many studies in multi-fidelity (MF) BO, *e.g.*, (Huang et al., 2006; Swersky et al., 2013; Lam et al., 2015; Picheny et al., 2013; Kandasamy et al., 2016, 2017a; Poloczek et al., 2017; McLeod et al., 2017; Wu and Frazier, 2017). While successful, these methods either ignore or oversimplify the strong, complex correlations between different fidelities, and hence might be inefficient in surrogate learning. For example, Picheny et al. (2013); Lam et al. (2015); Kandasamy et al. (2016); Poloczek et al. (2017) learned an independent GP for each fidelity; Song et al. (2019) used all the examples without discrimination to train one single GP; Huang et al. (2006); Takeno et al. (2019) imposed a linear correlation across fidelities, while Zhang et al. (2017) constructed a convolutional kernel as the cross-fidelity covariance and so the involved kernels in the convolution must be simple and smooth enough (yet less expressive) to obtain a closed form. Recently, Perrone et al. (2018) developed an NN-based multi-task BO method for hyper-parameter transfer learning. Their model constructs an NN feature mapping shared by all the tasks, and uses an independent linear combination of the mapped features to predict each task output. While we can consider each task as evaluating the objective at a particular fidelity, the model does not explicitly capture and exploit the correlations across different tasks — given the shared (latent) features, the predictions of these tasks (fidelities) are independent. The most recent work (Li et al., 2020) also developed an NN-based multi-fidelity BO method, which differs from our work in that (1) their model only estimates the relationship between successive fidelities, and hence has less capacity, (2) their work uses a recursive one-dimensional quadrature to calculate the acquisition function, and is difficult to extend to batch acquisitions. In a high level, the chain structure of Li et al. (2020)'s model also resembles deep GP based multi-fidelity models (Perdikaris et al., 2017; Cutajar et al., 2019).

Quite a few batch BO algorithms have been developed, such as (González et al., 2016; Wu and Frazier, 2016; Hernández-Lobato et al., 2017; Kandasamy et al., 2017b). However, they work with single-fidelity queries and are not easily extended to multi-fidelity optimization tasks. Takeno et al. (2019) proposed two batch querying strategies for their MF-BO framework. Both strategies

leverage the property that the covariance of a conditional Gaussian does not rely on the values of the conditioned variables; so, there is no need to worry about conditioning on function values that are still in query. The asynchronous version generates new queries conditioned on different sets of function values in query (asynchronously). However, if the conditional parts are significantly overlapping, which might not be uncommon in practice, there is a risk of generating redundant or even collapsed queries. Takeno et al. (2019) also talked about a synchronous version. While they discussed how to compute the information gain between the function maximum and a batch of function values, they did not provide an effective way to optimize it with the multi-fidelity querying costs. Instead, they suggested a simple heuristics to sequentially find each query by conditioning on the generated ones. However, there is no guarantee about this heuristics.

While in our experiments, we mainly use hyperparameter optimization to evaluate our multi-fidelity BO approach, there are many other excellent works specifically designed for hyperparameter tuning or selection, *e.g.*, the non-Bayesian, random search based method Hyberband (Li et al., 2017) which also reflects the multi-fidelity idea: it starts using few training iterations/epochs (low fidelity) to evaluate many candidates, rank them, iteratively selects the top-ranked ones, and further evaluate them with more iterations/epochs (high fidelity). BOHB (Falkner et al., 2018) is a hybrid of KDE based BO (Bergstra et al., 2011) and Hyperband. Li et al. (2018) further developed an asynchronous successive halving algorithm for parallel random search over hyperparameters. Domhan et al. (2015); Klein et al. (2017b) propose to estimate the learning curves, and early halt the evaluation of ominous hyperparameters according to the learning curve predictions. Swersky et al. (2014) introduced a kernel about the training steps, and developed Freeze-thaw BO (Swersky et al., 2014) that can temporarily pause the model training and explore several promising hyperparameter settings for a while and then continue on to the most promising one. The work in (Klein et al., 2017a) jointly estimates the cost as a function of the data size and training steps, which can be viewed as continuous fidelities, like in (Kandasamy et al., 2017a; Wu and Frazier, 2017).

# 6    Experiment

## 6.1    Surrogate Learning Performance

We first examined if BMBO-DARN can learn a more accurate surrogate of the objective. We used two popular benchmark functions: (1) *Levy* (Laguna and Martí, 2005) with two-fidelity evaluations, and (2) *Branin* (Forrester et al., 2008; Perdikaris et al., 2017) with three-fidelity evaluations. Throughout different fidelities are nonlinear/nonstationary transforms. We provide the details in the Appendix.

**Methods.** We compared with the following multi-fidelity learning models used in the state-of-the-art BO methods: (1) MF-GP-UCB (Kandasamy et al., 2016) that learns an independent GP for each fidelity. (2) MF-MES (Takeno et al., 2019) that uses a multi-output GP with a linear correlation structure across different outputs (fidelities), (3) Scalable Hyperparameter Transfer Learning (SHTL) (Perrone et al., 2018) that uses an NN to generate latent bases shared by all the tasks (fidelities) and predicts the output of each task with a linear combination of the bases. (4) Deep Neural Network Multi-Fidelity BO (DNN-MFBO) (Li et al., 2020) that uses a chain of NNs to model each fidelity, but only estimates the relationship between successive fidelities.

| *Levy* | nRMSE | MNLL |
|---|---|---|
| MF-GP-UCB | $0.831 \pm 0.195$ | $1.824 \pm 0.276$ |
| MF-MES | $0.581 \pm 0.032$ | $1.401 \pm 0.031$ |
| SHTL | $0.443 \pm 0.009$ | $1.208 \pm 0.026$ |
| DNN-MFBO | $0.365 \pm 0.035$ | $1.081 \pm 0.011$ |
| BMBO-DARN | $\mathbf{0.348 \pm 0.021}$ | $\mathbf{1.072 \pm 0.016}$ |
| *Branin* | | |
| MF-GP-UCB | $0.846 \pm 0.147$ | $1.976 \pm 0.208$ |
| MF-MES | $0.719 \pm 0.099$ | $1.796 \pm 0.128$ |
| SHTL | $0.835 \pm 0.218$ | $1.958 \pm 0.646$ |
| DNN-MFBO | $0.182 \pm 0.022$ | $0.973 \pm 0.013$ |
| BMBO-DARN | $\mathbf{0.158 \pm 0.016}$ | $\mathbf{0.965 \pm 0.005}$ |

Table 1: Surrogate learning performance on *Branin* function with three-fidelity training examples and *Levy* function with two-fidelity examples: normalized root-mean-square-error (nRMSE) and mean-negative-log-likelihood (MNLL). The results were averaged over five runs.

**Settings.** We implemented our model with PyTorch (Paszke et al., 2019) and HMC sampling based on the Hamiltorch library (Cobb and Jalaian, 2021) (`https://github.com/AdamCobb/hamiltorch`). For each fidelity, we used two hidden layers with 40 neurons and `tanh` activation. We ran HMC for 5K steps to reach burn in (by looking at the trace plots) and then produced 200 posterior samples with every 10 steps. To generate each sample proposal, we ran 10 leapfrog steps, and the step size was chosen as 0.012.

We implemented DNN-MFBO and SHTL with PyTorch as well. For DNN-MFBO, we used the same NN architecture as in BMBO-DARN for each fidelity, and ran HMC with the same setting for model estimation. For SHTL, we used two hidden layers with 40 neurons and an output layer with 32 neurons to generate the shared bases. We used ADAM (Kingma and Ba, 2014) to estimate the model parameters, and the learning rate was chosen from $\{10^{-4}, 5 \times 10^{-4}, 10^{-3}, 5 \times 10^{-3}, 10^{-2}\}$. We ran 1K epochs, which are enough for convergence. Note that we also attempted to use L-BFGS to train SHTL, but it often runs into numerical issues. ADAM is far more stable. We used a Python implementation of MF-MES and MF-GP-UCB, both of which use the RBF kernel (consistent with the original papers).

**Results**. We randomly generated $\{130, 65\}$ examples for *Levy* function at the two increasing fidelities, and $\{320, 130, 65\}$ examples for *Branin* function at its three increasing fidelities. After training, we examined the prediction accuracy of all the models with 100 test samples uniformly sampled from the input space. We calculated the normalized root-mean-square-error (nRMSE) and mean-negative-log-likelihood (MNLL). We repeated the experiment for 5 times, and report their average and standard deviations in Table. 1. As we can see, for both benchmark functions, BMBO-DARN outperforms all the competing models, confirming the advantage of our deep auto-regressive model in surrogate learning. Note that despite using a similar chain structure, DNN-MFBO is still inferior to BMBO-DARN, implying that our fully auto-regressive modeling (see (3)) can better estimate the relationships between the fidelities to facilitate surrogate estimation.

## 6.2 Real-World Applications

Next, we used BMBO-DARN to optimize the hyperparameters of four popular machine learning models: Convolutional Neural Networks (CNN) (Fukushima and Miyake, 1982; LeCun et al., 1990) for image classification, Online Latent Dirichlet Allocation (LDA) (Hoffman et al., 2010) for text mining, XGBoost (Chen and Guestrin, 2016) for diabetes diagnosis, and Physics-Informed Neural Networks (PINN) (Raissi et al., 2019) for solving partial differential equations (PDE).

**Methods and Setting.** We compared with the state-of-the-art multi-fidelity BO algorithms mentioned in Sec. 6.1, (1) MF-GP-UCB, (2) MF-MES, (3) SHTL, and (4) DNN-MFBO. In addition, we compared with (5) MF-MES-Batch (Takeno et al., 2019), the (asynchronous) parallel version of MF-MES, (6) SF-Batch (Kandasamy et al., 2017b) (`https://github.com/kirthevasank/gp-parallel-ts`), a single-fidelity GP-based BO that optimizes posterior samples of the objective function to obtain a batch of queries, (7) SMAC3 (`https://github.com/automl/SMAC3`), BO based on random forests, (8) Hyperband (Li et al., 2017) (`https://github.com/automl/HpBandSter`) that conducts multi-fidelity random search over the hyperparameters, (9) BOHB (Falkner et al., 2018) that uses Tree Parzen Estimator (TPE) (Bergstra et al., 2011) to generate hyperparameter candidates in Hyperband iterations. We also tested our method that queries at one input and fidelity each time ($B = 1$), which we denote by BMBO-DARN-1. We used the same setting as in Sec. 6.1 for all the multi-fidelity methods, except that for SHTL, we ran 2K epochs in surrogate training to ensure the convergence. For our method, we set the maximum number of iterations in optimizing the batch acquisition function (see Algorithm 8) to 100 and tolerance level to $10^{-3}$. For the remaining methods, *e.g.*, SMAC3 and Hyperband, we used their original implementations and default settings. For all the batch querying methods, we set the batch size to 5. All the single fidelity methods queried at the highest fidelity.

**Convolutional Neural Network (CNN).** Our first application is to train a CNN for image classification. We used CIFAR-10 dataset (`https://www.cs.toronto.edu/~kriz/cifar.html`), from which we used 10K images for training and another 10K for evaluation. To optimize the hyperparameters, we considered three fidelities, *i.e.*, training with 1, 10, 50 epochs. We used the average negative log-loss (nLL) to evaluate the prediction accuracy of each method. We considered optimizing the following hyperparameters: # convolutional layers ranging from [1,4], # channels in the first filter ([8, 136]), depth of the dense layers ([1, 8]), width of the dense layers ([32, 2080]), pooling type (["max", "average"]), and dropout rate ([$10^{-3}$, 0.99]). We optimized the dropout rate in the log domain, and used a continuous relaxation of the discrete parameters.

Initially, we queried at 10 random hyperparameter settings at each fidelity. All the methods started with these evaluation results and repeatedly identified new hyperparameters. We used the average running time at each training fidelity as the cost: $\lambda_1 : \lambda_2 : \lambda_3 = 1 : 10 : 50$. After each query, *we evaluated the performance of the new hyperparameters at the highest*

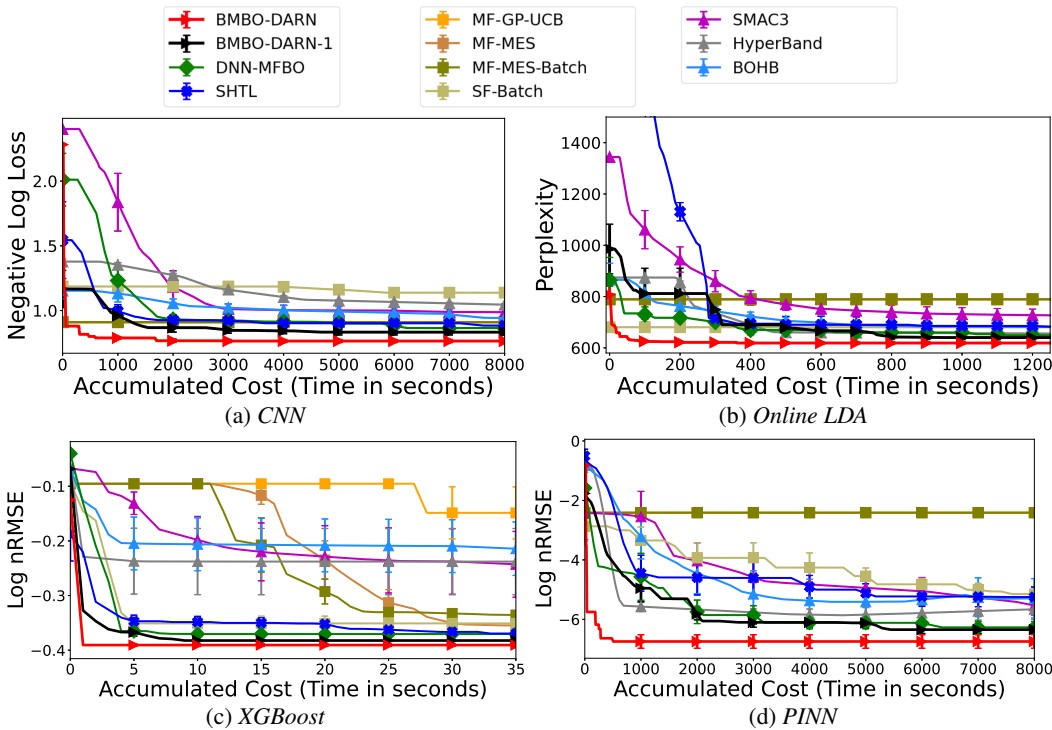

(a) *CNN*  (b) *Online LDA*

(c) *XGBoost*  (d) *PINN*

Figure 2: Performance *vs.* accumulated cost (running time) in Hyperparameter optimization tasks. For fairness, all the batch methods queried new examples sequentially, *i.e.*, no parallel querying was employed. The results were averaged over five runs. Note that MF-GP-UCB, MF-MES and MF-MES-Batch often obtained very close results and their curves overlap much.

*training level*. We ran each method until 100 queries were issued. We repeated the experiment for 5 times and in Fig. 2a report the average accuracy (nLL) and its standard deviation for the hyperparameters found by each method throughout the optimization procedure.

**Online Latent Dirichlet Allocation (LDA).**
Our second task is to train online LDA (Hoffman et al., 2010) to extract topics from 20NewsGroups corpus (http://qwone.com/~jason/20Newsgroups/). We used 5K documents for training, and 2K for evaluation. We used the implement from the scikit-learn library (https://scikit-learn.org/stable/). We considered optimizing the following hyperparameters: document topic prior $\alpha \in [10^{-3}, 1]$, topic word prior $\eta \in [10^{-3}, 1]$, learning decay $\kappa \in [0.51, 1]$, learning offset

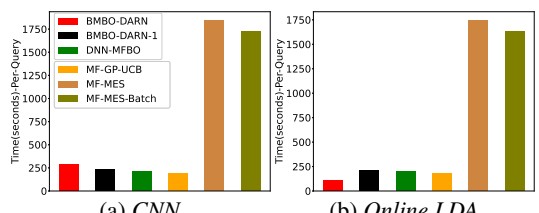

(a) *CNN*  (b) *Online LDA*

Figure 3: Average time to generate queries (including surrogate training).

$\tau_0 \in [1, 2, 5, 10, 20, 50, 100, 200]$, E-step stopping tolerance $\epsilon \in [10^{-5}, 10^{-1}]$, document batch size in $[2, 4, 8, 16, 32, 64, 128, 256]$, and topic number $K \in [1, 64]$. We optimized $\alpha, \eta, \kappa$ and $\epsilon$ in the log domain, and used a continuous relaxation of the discrete parameters. We considered three fidelities — training with 1, 10 and 50 epochs, and randomly queried 10 examples at each fidelity to start each method. We evaluated the performance of the selected hyperparameters in terms of perplexity (the smaller, the better). In Fig. 2 b, we reported the average perplexity (and its standard deviation) of each method after five runs of the hyperparameter optimization.

**XGBoost.** Third, we trained an XGBoost model (Chen and Guestrin, 2016) to predict a quantitative measure of the diabetes progression (https://archive.ics.uci.edu/ml/datasets/diabetes). The dataset includes 442 examples. We used two-thirds for training and the remaining one-third for evaluation. We used the implementation from the scikit-learn library. We optimized the following hyperparameters: Huber loss parameter $\alpha \in [0.01, 0.1]$, the non-negative complexity pruning parameter ($[0.01, 100]$), fraction of samples used to fit individual base learners ($[0.1, 1]$),

fraction of features considered to split the tree ($[0.01, 1]$), splitting criterion (["MAE", "MSE"]), minimum number of samples required to split an internal node ($[2, 9]$), and the maximum depth of individual trees ($[1, 16]$). The hyperparameter space is 12 dimensional. We considered three fidelities — training XGBoost with 2, 10 and 100 weak learners (trees). The querying cost is therefore $\lambda_1 : \lambda_2 : \lambda_3 = 1 : 5 : 50$. We started with 10 random queries at each fidelity. We used the log of nRMSE to evaluate the performance. We ran 5 times and report the average log-nRMSE of the identified hyperparameters by each method in Fig. 2c.

**Physics-informed Neural Networks (PINN).** Our fourth application is to learn a PINN to solve PDEs (Raissi et al., 2019). The key idea of PINN is to use boundary points to construct the training loss, and meanwhile use a set of collocation points in the domain to regularize the NN solver to respect the PDE. With appropriate choices of hyperparameters, PINNs can obtain very accurate solutions. We used PINNs to solve Burger's equation (Morton and Mayers, 2005) with the viscosity $0.01/\pi$. The solution becomes sharper with bigger time variables (see the Appendix) and hence the learning is quite challenging. We followed (Raissi et al., 2019) to use fully connected networks and L-BFGS for training. The hyperparameters include NN depth ($[1, 8]$), width ($[1, 64]$), and activations (8 choices: `Relu`, `tanh`, `sigmoid`, their variants, *etc*.). Following (Raissi et al., 2019), we used 100 boundary points as the training set and 10K collocation points for regularization. We used 10K points for evaluation. We chose 3 training fidelities, running L-BFGS with 10, 100, 50K maximum iterations. The querying cost (average training time) is $\lambda_1 : \lambda_2 : \lambda_3 = 1 : 10 : 50$. Note that in fidelity 3, L-BFGS usually converged before running 50K iterations. We initially issued 10 random queries at each fidelity. We ran each method for 5 times and reported the average log nRMSE after each step in Fig. 2d.

**Results**. As we can see, in all the applications, BMBO-DARN used the smallest cost (*i.e.*, running time) to find the hyperparameters that gives the best learning performance. In general, BMBO-DARN identified better hyperparameters with the same cost, or equally good hyperparameters with the smallest cost. BMBO-DARN-1 outperformed all the one-by-one querying methods, except that for online LDA (Fig. 2b) and PINN (Fig. 2d), it was worse than DNN-MFBO and Hyperband at the early stage, but finally obtained better learning performance. We observed that the GP based baselines (MF-MES, MF-GP-UCB, SF-Batch, *etc*.) are often easier to be stuck in suboptimal hyperparameters, this might because these models are not effective enough to integrate information of multiple fidelities to obtain a good surrogate. Together these results have shown the advantage of our method, especially in our batch querying strategy. Finally, we show the average query generation time of BMBO-DARN for CNN and Online LDA in Fig. 3 (including surrogate training). It turns out BMBO-DARN spends much less time than MF-MES using the global optimization method DIRECT (Jones et al., 1998), and comparable to MF-GP-UCB and DNN-MFBO. Therefore, BMBO-DARN is efficient to update the surrogate model and generate new queries.

## 7 Conclusion

We have presented BMBO-DARN, a batch multi-fidelity Bayesian optimization method. Our deep auto-regressive model can serve as a better surrogate of the black-box objective. Our batch querying method not only is efficient, avoiding combinatorial search over discrete fidelities, but also significantly reduces the cost while improving the optimization performance.

## Acknowledgments

This work has been supported by MURI AFOSR grant FA9550-20-1-0358.

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
