# Appendix

# 1 Synthetic Benchmark Functions

## 1.1 Branin Function

The input is two dimensional, $\mathbf{x} = [x_1, x_2] \in [-5, 10] \times [0, 15]$. We have three fidelities to evaluate the function, which, from high to low, are given by

$$
\begin{aligned}
f_3(\mathbf{x}) &= -\left(\frac{-1.275x_1^2}{\pi^2} + \frac{5x_1}{\pi} + x_2 - 6\right)^2 - \left(10 - \frac{5}{4\pi}\right)\cos(x_1) - 10, \\
f_2(\mathbf{x}) &= -10\sqrt{-f_3(x-2)} - 2(x_1 - 0.5) + 3(3x_2 - 1) + 1, \\
f_1(\mathbf{x}) &= -f_2\big(1.2(\mathbf{x}+2)\big) + 3x_2 - 1.
\end{aligned}
\tag{1}
$$

We can see that between fidelities are nonlinear transformations, nonuniform scaling, and shifts.

## 1.2 Levy Function

The input is two dimensional, $\mathbf{x} = [x_1, x_2] \in [-10, 10]^2$. We have two fidelities,

$$
\begin{aligned}
f_2(\mathbf{x}) &= -\sin^2(3\pi x_1) - (x_1 - 1)^2[1 + \sin^2(3\pi x_2)] - (x_2 - 1)^2[1 + \sin^2(2\pi x_2)], \\
f_1(\mathbf{x}) &= -\sqrt{1 + f_2^2(\mathbf{x})}.
\end{aligned}
\tag{2}
$$

# 2 Details about Physics Informed Neural Networks

Burgers' equation is a canonical nonlinear hyperbolic PDE, and widely used to characterize a variety of physical phenomena, such as nonlinear acoustics (Sugimoto, 1991), fluid dynamics (Chung, 2010), and traffic flows (Nagel, 1996). Since the solution can develop discontinuities (i.e., shock waves) based on a normal conservation equation, Burger's equation is often used as a nontrivial benchmark test for numerical solvers and surrogate models (Kutluay et al., 1999; Shah et al., 2017; Raissi et al., 2017).

We used physics informed neural networks (PINN) to solve the viscosity version of Burger's equation,

$$
\frac{\partial u}{\partial t} + u\frac{\partial u}{\partial x} = \nu\frac{\partial^2 u}{\partial x^2},
\tag{3}
$$

where $u$ is the volume, $x$ is the spatial location, $t$ is the time, and $\nu$ is the viscosity. Note that the smaller $\nu$, the sharper the solution of $u$. In our experiment, we set $\nu = \frac{0.01}{\pi}$, $x \in [-1, 1]$, and $t \in [0, 1]$. The boundary condition is given by

$$
u(0, x) = -\sin(\pi x), \quad u(t, -1) = u(t, 1) = 0.
$$

We use an NN $u_{\mathcal{W}}$ to represent the solution. To estimate the NN, we collected $N$ training points in the boundary, $\mathcal{D} = \{(t_i, x_i, u_i)\}_{i=1}^N$, and $M$ collocation (input) points in the domain, $\mathcal{C} = \{(\hat{t}_i, \hat{x}_i)\}_{i=1}^M$. We then minimize the following loss function to estimate $u_{\mathcal{W}}$,

$$
L(\mathcal{W}) = \frac{1}{N}\sum_{i=1}^N (u_{\mathcal{W}}(t_i, x_i) - u_i)^2 + \frac{1}{M}\sum_{i=1}^M \left(\big|\psi(u_{\mathcal{W}})(\hat{t}_i, \hat{x}_i)\big|^2\right),
$$

where $\psi(\cdot)$ is a functional constructed from the PDE,

$$\psi(u) = \frac{\partial u}{\partial t} + u\frac{\partial u}{\partial x} - \nu\frac{\partial^2 u}{\partial x^2}.$$

Obviously, the loss consists of two terms, one is the training loss, and the other is a regularization term that enforces the NN solution to respect the PDE.