# OpenReview forum: "Batch Multi-Fidelity Bayesian Optimization with  Deep Auto-Regressive Networks"
_NeurIPS.cc/2021/Conference — NeurIPS 2021 Poster_

### Official Review · Reviewer_gu5R · 2021-07-12

**Rating:** 7
**Confidence:** 4

**Summary:**

Multi-fidelity methods are prevalent in problem settings where high-quality (but difficult to acquire) observations are enhanced with lower-quality (but more easily obtained) samples of a function. In this paper, the authors focus on the application of multi-fidelity modelling to hyper parameter tuning, whereby different model settings are verified with varying fidelity constraints (such as epochs in a training loop) in order to achieve the best accuracy-cost trade-off. The model introduced here is based on chained neural networks, whereby the input at each fidelity layer combines the input in the original domain for the function to be estimated, along with the outputs from the preceding fidelities. In order to carry out BO, the authors develop a batch acquisition technique that encourages diversity across the samples being drawn. This is in contrast to other techniques where new samples are drawn one a time, thus incurring the risk of there not being sufficient diversity across successive iterations. In order to achieve this goal, the authors also propose an alternating optimisation scheme for iteratively selecting samples across different fidelities.

**Limitations And Societal Impact:**

As highlighted in the main review, I think it would interesting for the authors to expose the limitations for this work more clearly. Having more insight into what could constitute future work would also be appreciated. There is no immediate societal impact to consider here.

**Main Review:**

### Strengths

- Having a multi-fidelity model that can be trained end-to-end is an appealing property, as it allows for learning the interactions across fidelities in a unified manner rather than disjointed. The batch acquisition function also arises quite naturally from the formulation of the model, meaning that the authors do not have to rely on a multitude of empirical tricks and implementation hacks to meet the desired goal.

- The performance of the method is evaluated over over benchmark functions (for assessing the quality of the multi fidelity model in isolation), while four hyper-parameter optimisation set-ups are used to validate the practical effectiveness of the BO proposal. This variety of experiments successfully addresses the different potential uses that the methods proposed in the paper might have.

- The paper is very well-written and organised. The literature review adequately covers related work, and the experimental evaluation also includes a wide variety of competing methods, some of which were implemented by the authors themselves.


### Weaknesses

- The importance of having a multi-fidelity model that is either chained or trained end-to-end has already been explored by Perdikaris et al (2017) and Cutajar et al (2019) for deep GP model constructions. I think that discussing this connection could be beneficial to the paper as the superior uncertainty estimates that are typically expected of GP models should be relevant to the iterative BO procedure explored here.

- It would be interesting to more explicitly state the potential limitations of the proposal here. For example, how well does the MF model scale to large input dimensions, either in terms of computational efficiency or expressivity? The paper successfully leverages weaknesses in other methods for motivating the work work described here, but there is little insight into which conditions might cause this method to underperform.

- In the same vein of the above, the paper doesn’t really hint at how the method described here can be improved or extended further in future work.


### Writing and Clarity

The paper is very well-written and a pleasure to read. The proposal is well-motivated and the multi-fidelity batch acquisition function is properly explained. One possible suggestion would be to include a visualisation of the multi-fidelity model itself (and how the output of the various layers feeds into the next) within the main paper itself.

Some very minor typos:
- *L2:* between *the* cost and accuracy
- *L15:* reference to *alternating optimization* is a bit confusing here.
- *L41:* between successive queries *and* is at higher
- *L83:* most commonly used *class of surrogate models are Gaussian processes*
- *L114* might limit the *expressivity*
- *References:* Certain terms (e.g. Bayesian) and conference names need to appear as capitalised in the references.


### Overall Recommendation

Besides putting forward an interesting problem to solve, the authors propose a well-rounded solution and provide multiple experiments that showcase the diversity of settings this work can be applied to. Just as importantly, the paper and its writing are already in a state fit for publication, which gives me greater confidence in recommending it for acceptance.

**Time Spent Reviewing:**

15

---

> ### Author Response · Authors · 2021-08-10
> **To Reviewer 4(gu5R):**
>
> Thanks for your detailed and valuable comments. Here are our responses.
>
> C: comment; R: response
>
> C1: The importance of having a multi-fidelity model that is either chained or trained end-to-end has already been explored by Perdikaris et al (2017) and Cutajar et al (2019) for deep GP model constructions. I think that discussing this connection could be beneficial to the paper as the superior uncertainty estimates that are typically expected of GP models should be relevant to the iterative BO procedure explored here
>
> R1: We will elaborate on the connection between our model and deep DP in the related work for surrogate modeling.
>
> C2: For example, how well does the MF model scale to large input dimensions, either in terms of computational efficiency or expressivity? The paper successfully leverages weaknesses in other methods for motivating the work wodescribed here, but there is little insight into which conditions might cause this method to underperform.
>
> R2:  We present the surrogate model performance comparison in Table 1 shows that our model is more expressive than other surrogate models. Intuitively, NN surrogates usually are more efficient for handling high-dimensional input [1]. However, the uncertainty estimation of deep models is still an area not fully explored. Compared to GP’s native uncertainty nature, the NN surrogate might not be a good choice when the hyper-parameters space is small, and fidelity levels are not significantly distinguished. In that case, a single GP on the highest fidelity would be sufficient.
>
> C3: the paper doesn’t really hint at how the method described here can be improved or extended further in future work.
>
> R3: First, although HMC generates high-quality samples for applications like XGBoost, training the surrogate takes longer than the tasks themselves. Improve the inference efficiency could be an immediate future work. Second, what if the cost is not aware by the acquisition function? This is a common scenario when the fidelities parameters do no reflects the actual costs.
>
> [1] Snoek, J., Rippel, O., Swersky, K., Kiros, R., Satish, N., Sundaram, N., ... & Adams, R. (2015, June). Scalable bayesian optimization using deep neural networks. In International conference on machine learning (pp. 2171-2180). PMLR.

---

> > ### Comment · Reviewer_gu5R · 2021-08-31
> > **Acknowledgement of Author Rebuttal**
> >
> > Thank you for your response - my score for the paper remains the same. I have also read through all other reviews/rebuttals for this submission and am participating in the discussion on the paper's outcome.

---

### Official Review · Reviewer_Hcby · 2021-07-14

**Rating:** 6
**Confidence:** 4

**Summary:**

In multi-fidelity black-box optimization, f() can be queried at points x at different levels of precision. Higher levels of precision have higher costs. Prior work on Bayesopt for mfbbo used GP models for multi-fidelity observations that make simplistic assumptions about correlations across fidelities. In response, the authors propose instead using a flexible autoregressive model for modeling the sequence of fidelity scores for a given x. Bayesian inference on the model is performed using HMC.

The paper also introduces a batched form of max-value entropy search and an algorithm for optimizing this complex acquisition function that works well in practice.

The experimental results are impressive. The proposed method outperforms a large number of popular methods from the literature.

**Main Review:**

=originality=
The paper provides multiple novel methods (surrogate model, batch acquisition function, etc) that will be of broader interest to the Bayesopt community.

=clarity=
Overall, good. See comment below about the problem statement, though.

=significance=
Multi-fidelity black-box optimization has applications in a number of important domains, including neural network hyper-parameter tuning. This paper benchmarks against a variety of recent algorithms from the literature and demonstrates an impressive performance improvement.

=comments=
The paper would be improved by a more formal problem statement. Lines 92-106 are largely focused on a discussion of related work. It's unclear to me, for example, why eq (2) is necessary. It would be helpful to better explain the action space for the algorithm. For example, do fidelities have to be queried in increasing order of cost? If you query f() at high fidelity, does that mean that you get a low-fidelity measurement for free? This is the case for neural network training (where high fidelity corresponds to more training steps), but not true in general.

 I'm assuming that for each x you can evaluate f at any given fidelity. However, the parametrization of the autoregressive model in (3) suggests that predicting at a given fidelity requires having data for every available measurement of x at lower fidelity. This places a considerable restriction at the action space that isn't reflected in a problem definition or in sec 4.3. There are ways to parametrize an autoregressive model such that it can handle missing data, but this is not discussed.

Also, it was unclear to me why you need to have a separate model for each fidelity in (4). Wouldn't there be a shared model that could model all of the fidelities, and thus would have more training data?

I'm not that familiar with the latest literature on max-value entropy search. Can you comment on what is novel in your usage of it. It was unclear to me, for example, what parts of sections 4.1 and 4.2 are new.

The experiments section contains a thorough ablation analysis of the impact of various surrogate models. This helps provide evidence for the importance of one of the paper's contributions. However, the paper's other contributions (batch acquisition function, strategy for optimizing the AF) are not analyzed. I'm curious, for example, what would happen if you used a simpler acquisition function that doesn't account for interactions within the batch or used something other than MVES.

**Time Spent Reviewing:**

2

---

> ### Author Response · Authors · 2021-08-10
> **To Reviewer 3(Hcby):**
>
> Thanks for your detailed and valuable comments. Here are our responses.
>
> C: comment; R: response
>
> C1: The paper would be improved by a more formal problem statement. Lines 92-106 are largely focused on a discussion of related work. It's unclear to me, for example, why eq (2) is necessary. It would be helpful to better explain the action space for the algorithm. For example, do fidelities have to be queried in increasing order of cost? If you query f() at high fidelity, does that mean that you get a low-fidelity measurement for free? This is the case for neural network training (where high fidelity corresponds to more training steps), but not true in general.
>
> R1: Thank you for your suggestion; we will consider how to make the related work more elaborative and highlight our contribution. We will try to make it more elaborative and highlight the contributions.  We don’t hold any assumptions about the query order. For an iterative algorithm, when you have the high-fidelity sample, then the low-fidelity sample is free indeed!  However, we want to maintain the query in a black-box way. You will not get the extras that you have asked for. Because, for other tasks, the benefits have shown in iterative algorithms do not exist. So we still want to make it task agnostic.
>
> C2: suggests that predicting at a given fidelity requires having data for every available measurement of x at lower fidelity
>
> R2: No, our model does not require low-fidelity measurements to make high-fidelity predictions. What (3) suggests is that the high-fidelity output relies on all previous low-fidelity outputs as input. Note that f_m is the output of NN and y_m is the measurement; we have distinguished them in our formulation.
>
> C3: Can you comment on what is novel in your usage of it. It was unclear to me, for example, what parts of sections 4.1 and 4.2 are new.
>
> R3: First, we use a full-autoregressive that high-fidelity takes all previous fidelity outputs. Second, in section 4.1, we propose the batch acquisition function for multi-fidelity BO. Why batch queries matter in BO except for parallelism, please check R1 to reviewer 1. Third, in section 4.2, we present a tractable computation of this batch acquisition. Note that find the best batch to optimize the batch mutual information cannot be resolved in polynomial time. So, in 4.3, we propose to use an alternative ascending update to find the near-optimal batch.
>
> C4: However, the paper's other contributions (batch acquisition function, strategy for optimizing the AF) are not analyzed. I'm curious, for example, what would happen if you used a simpler acquisition function that doesn't account for interactions within the batch or used something other than MVES
>
> R4: In our experiments, we explored several BO methods with variate surrogate models and variate acquisition functions. BMBO-DARN-1 is our method with a sequential query strategy that rules out the impact of the batch. DNN-MFBO uses a similar autoregressive model but uses moment-matching propagation as inference instead of HMC. SHTL models the fidelities in a shared base fashion with EI as a more straightforward acquisition. MF-MES, MF-MES-Batch uses linear correlated GPs as surrogate, MES as acquisition, and MF-GP-UCB uses independent GPs and UCB as acquisition.

---

### Official Review · Reviewer_zV6Z · 2021-07-16

**Rating:** 4
**Confidence:** 4

**Summary:**

The paper suggests a methodology for performing multi-fidelity Bayesian Optimziation (BO) using a particular (deep auto-regressive) Neural Network model as the surrogate. The authors formulate a batch acquisition strategy based on max-value entropy search, and propose an approximation using posterior sampling and cyclic optimization that renders the optimization of the acquisition function tractable. The method is evaluated on both synthetic and real experiments, where it shows improved performance over existing approaches.

**Limitations And Societal Impact:**

-There is more room for discussing limitations; see main review for details.
- No concerns regarding social impact.



**Main Review:**


### High-level summary
- The problem setting is relevant and of interest to the community.
- The modeling approach seems quite an incremental contribution over previous work, and it is unclear what the effect of the model on the performance of the overall approach is.
- There is some but not much methodological novelty in the approach; it essentially uses a standard batch MES formulation with a (slightly modified compared to previous work) BNN model and  some approximations of the acquisition function to render computations tractable. The main novelty is in these approximations; however, I am not convinced of the merits suggested approach.
- The writing is generally ok (some wording issues, repetitive in some places), overall the paper is relatively easy to follow.
- The empirical evaluation includes a good number of baselines, but is quite sparse on details and thus not clear; this section needs to be improved. In the current form, it's hard to assess the significance of the results.
- Overall, a potentially interesting approach, but too many open questions in the current draft.
- Not making the bar for NeurIPS in the current form.


### Notes
- The authors appear to suggest that all GP approaches have computational complexity of N^3: There area  number of of ways around this (e.g. linearge conjugate gradient solves instead of Cholesky-based approaches, interpolation / sparse GPs, variational GPs, etc.). These have some other challenges but they should at least be mentioned as ways to reduce computational complexity.
- In all the examples considered in the paper, the different fidelities correspond to different "lengths" (epochs, weak learners, LBFGS-B steps). An alternative approach to pre-committing to the length of the evalution is to to not specify the fidelity of a candiate priori but perform early stopping while obtaining partial results. Such approaches are at least as good as pre-committing (since that is a special case of the early stopping setting). The paper should discuss the early stopping line of work; please also explain why you consider the pre-comitting approach.
- Modeling:
  - The modeling approach is not particularly well motivated and the contribution seems quite incremental compared to previous work.
  - Specifically, the main difference to the model of Li et al. is that fidelities here depend on the output of all previously modeled fidelities. This doesn't really cause any technical challenges and is a reasonably setup, but it's not clear from the paper what the effect of this change is.
  - In particular, there is no evaluation of the model fits, and a lack of an ablation study comparing to the simpler model from Li et al. means that it's not possible to separate the effect of the model and the effect of the different approach to optimizing the acquisition function (in the case of sequential optimziation).
  - This point is particular pertinent since in all of the real-world examples considered, one would expect a relatively simple (and not complex) relationship between the fidelities. It's not clear that in any of the real world the examples the additional model capacity obtained by the modified structure would be all that helpful.
  - The model is motivated with "complex, strong (nonstationary, nighly nonlinear) correlations across fidelites": Can you give some more conrete examples to provide some motiviation/intuition for where/in which practical applications this is / would be the case?
  - Model fitting fo BNNs is known to be much more finnicky than other approaches. The authors state they fit models by "looking at the trace plots", which does not seem particularly robust / automatable (e.g. "step size was chosen as 0.012" - why and how? how hard was it to determine this value?)
- Acquisition Function:
  - Sec 4.1 is a bit light on discussing similarities and differnces to other batch acquisition functions. How does this approach relate to the asynchronous variant in Takeno et al.? It would also be useful to discuss the relation to the GIBBON batch MVES approximation developed in [1].
  - There seems to be a bit of a tension between the modelinga approach using a BNN (much more flexible than existing GP-based formulations) and the efficient computation scheme that is based on moment moment matching.
    - The approach is based on Laplace approximation, which is presumably a pretty coarse approximation. If that is true, what is the motivation for building a complicated BNN model in the first place? It seems that the cases in which the more expressive model helps would likely be those where the approximation used in the computation of the acquisition function will be poor.
    - It's not clear how well the Laplace approximation works for this problem. It woudl be helpful to verify in simulations on real problems that this is in fact a reasonable thing to do.
  - The coordinate descent approach optimization is reasonable but has obvious issues that should be discussed (e.g. this can get easily get stuck in local optima).
- Experiments:
    - The difference between BMBO-DARN and DNN-MFBO isn't particularly large (would be good to see p-values of a hypothesis test for the difference).
    - The real-world application evaluation seems to be quite comprehensive, a few models and quite a few baselines. However, additional ablation studies are necessary to understand the impact of the modeling change and the impact of the acquisition function formulation (see above).
    - There is lack of detail on the evaluation criteria. Specifically, it is not clear to me what exactly the accumulated cost is, and how it is computed in the batch setting (e.g. how is it defined if there are heterogeneous fidelities that take different times to evaluate?). Reading this section, I was left confused and concerned that there might be an apples-to-oranges comparison of non-batch vs. batch methods going on. IN particular, the fact that BMBO-DARN-1 is a varaint that "queries at one input and fidelity each time (B = 1)" performs worse thatn BMBO-DARN is surprising, since purely sequential optimization should perform much better than any batch variant if given the same fidelities to choose from. Essentially I have two ways to read this: (1) The cost function is not actually summed across the cost of the batches running in parallel (i.e. only overall wall time is counted, not total CPU time), or (2) the BMBO-DARN-1 only evalues at the highest fidelity. EIther version would be somewhat misleading (in the latter case an obvious baseline that would need to be added would be one that only allows low fidelity evaluations to make sure the improvements observed here don't just come from the fact that using lower fidelities is generally enough). Please clarify this and be explicit about the evaluation metrics and the benchmark procedure.
    - Why don't the losses all start at the same value if the initial points are shared between methods?
    - How many function evaluations are actually performed in the experiments?
- Miscellaneous:
    - Somewhat repetitive discussion of "other approaches simplifying things" (introduction, background, related work): should consolidate this somehow as the repeated discussion does't really add anything
- Wording (selected):
    - "most commonly used surrogate is GP"
    - "this might because these models are not effective"



### Questions
- You mention "the standard one by one querying strategy": Can you elaborate on this? I don't think this is true, there has been a lot of work on batch and fully asynchronous candidate generation in the past (see e.g. ), e.g. via sequential conditioning on pending points.
- "running L-BFGS with 10, 100, 50K maximum iterations": Is this correct? this seems like an obscene difference in fidelities. It would also seem odd with the average training time. Or is the idea that 50K basically just means "LBFGS until convergence"?
- Timings in Figure 2:
    - Is this the average time per candidate or the average time per batch?
    - How much time is spent on model fitting, how much on acqf optimization?
- What MF-MES implementation is used? The paper just states "We used a Python implementation of MF-MES and MF-GP-UCB". It seems from Figure 2 that whatever was used is extremely slow; it seems quite possible to substantially reduce this time (and likely also optimization performance) with a proper implementation.
- Is there an assumption that the fidelities are ordered? If so, what if that's not the case? Will things still work ok? It seems like the auto-regressive model might not be the best approach in that case.
  - How could this approach work with continuous fidelities? These are common e.g. in HPO for ML training;  incorporating some prior information about the structure information should help with the modeling.


### References
[1] Moss, M., et al., GIBBON: General-purpose Information-Based Bayesian Optimisation. arXiv:2102.03324, 2020


**Time Spent Reviewing:**

2.5

---

> ### Author Response · Authors · 2021-08-10
> **To Reviewer 2(zV6Z):**
>
> C: comment; R: response
>
> C1: The authors appear to *suggest that all GP approaches have computational complexity of N^3*: There area number of of ways around this (e.g. linearge conjugate gradient solves instead of Cholesky-based approaches, interpolation / sparse GPs, variational GPs, etc.).
>
>
> R1: We **never** criticize or “suggest that all GP approaches have computational complexity of N^3”. O(N^3) only appears in Line 208-211 in Related Work, where we discuss the seminal work of Sonek et al. 2015 in BNN based BO and mention the advantage of their method, namely, the linear scalability in training. Obviously, here O(N^3) means the standard/exact GP inference, which is used by most GP-BO algorithms. Note that this advantage is also stressed on in the original paper (Sonek et al. 2015). We would love to supplement the discussion about sparse GP approximation, etc.
>
> C2: The paper should discuss the early stopping line of work; please also explain why you consider the pre-comitting approach.
>
> R2:  Our topic/work is multi-fidelity batch BO and is irrelevant to early-stopping. Regarding the experimental setting, first, we evaluate our work with hyper-parameter optimization, which is a very popular type of BO tasks (appeared in many, many BO papers). Second, the multi-fidelity settings, namely, rough and full training (according to epochs/iterations), are common in practically tuning the hyperparameters. For example, the well-known Hyperband(HB) and BOHB also use the same the multi-fidelity ideas to reduce the cost and tune the hyperparameters (see Line 237-241). We systematically compared with them in the experiments. Third, we also cite and discuss many other works related to hyper-parameter tuning, see Line 235-249.
>
> C3: “the contribution seems quite incremental compared to previous work” (Li et al. 2020)
>
> R3: First, from the modeling perspective, as stated in our paper, our model is a “full auto-regressive model” and can capture the relationship between each fidelity and all the prior fidelities. The model by Li et al. 2020 only captures the relationship of successive fidelities. Hence, our model is more expressive. In the ablation study, our model indeed achieves better prediction accuracy than (Li et al. 2020). See Table 1.
>
> Second, in model estimation, Li et al. 2020’s method, “sequential, fidelity-wise Gauss-Hermite quadrature and moment-matching” to estimate the posterior of the function output is no longer applicable to our model, because the quadrature dimension is increased, and it becomes extremely complex and costly. Instead, we used HMC for inference and posterior samples to construct a joint Gaussian distribution, which is completely different.
>
> Third, our major contribution, i.e., batch querying, has never shown up in (Li et al. 2020), not to mention “incremental”.  We first extend the single-query acquisition function to a batch multi-fidelity version, which explicitly considers the correlation within the query batch, penalize high-correlations, encourages diversity and account for the costs (see Line 156-162).  Therefore, our method can effectively avoid the sample collapse or redundance issue. Next, our method of computing the acquisition function based on empirical covariance matrices and join entropy is simple and efficient and has nothing to do with (Li et al. 2020) that uses quadrature and conditional entropy approximation. We then develop an efficient method to maximize the batch acquisition function (i.e., fidelity-wise alternating optimization). The complexity of our optimization is linear with the number of fidelities. This is NOT trivial, because the straightforward maximalization incurs a combinatorial search, and the complexity is exponential to the number of fidelities; It will extremely costly (even infeasible) for moderate batches (e.g., 10 or 20), not to mention larger ones.
>
> Therefore, our method differs from (Li et al. 2020) substantially in all aspects: modeling, inference, and acquisition function calculation and optimization. We do NOT believe these contributions are “quite incremental compared to previous work”.
>
> C4: …*no evaluation of the model fits, and a lack of an ablation study comparing to the simpler model from Li et al*. means that it's not possible to separate the effect of the model and the effect of the different approach to optimizing the acquisition function
>
> R4: **We DID compare with “simpler model from Li et al. 2020 in an ablation study”. See Sec 6.1 and Table 1.** We compared the surrogate learning performance on small samples and two synthetic benchmarks. The results demonstrates the advantage of our method in surrogate learning, which is from “the effect of the model”.
>
> C5: “in all of the real-world examples considered, one would expect a relatively simple (and not complex) relationship between the fidelities…. Can you give some more conrete examples to provide some motiviation/intuition for where/in which practical applications this is / would be the case?”
>
> R5: We do NOT believe “a relatively simple relationship between the fidelities” is realistic in practice, given real-world applications are so complex and diverse. Many works have confirmed across the fidelities are often strong, complex (nonlinear) relationships in practical applications. For example, [1][2] used GPs with nonlinear kernels to model the fidelity relationships and outperforms linear auto-regressive models **in predicting mixed convection flows and sequential experiment design to fetch new observations of infection Rates of Plasmodium falciparum among African children. [3]** developed deep matrix GPs to model multiple fidelities for high-dimensional outputs, which is much superior to baselines **in PDE solving and fluid dynamics. [4]** proposed a multi-resolution GP model based on Gaussian process regression networks and mixture of experts (where is the fidelity correlation is nonstationary and nonlinear) and demonstrates the advantage **in sensor networks (fidelities are represented by the sensor resolutions)**. [5,6] investigate and try to predict Learning curves in machine learning applications, and show that **the learning curves are typically nonlinear**, confirming that our fidelities in the experiments (performance at different epochs/iterations) are nonlinear correlated in nature.
>
>
> [1] Perdikaris, P., Raissi, M., Damianou, A., Lawrence, N. D., & Karniadakis, G. E. (2017). Nonlinear information fusion algorithms for data-efficient multi-fidelity modelling. Proceedings of the Royal Society A: Mathematical, Physical and Engineering Sciences, 473(2198), 20160751.
>
> [2] Cutajar, Kurt, et al. "Deep gaussian processes for multi-fidelity modeling." arXiv preprint arXiv:1903.07320 (2019).
>
> [3] Wang, Zheng, et al. "Multi-Fidelity High-Order Gaussian Processes for Physical Simulation." International Conference on Artificial Intelligence and Statistics. PMLR, 2021.
>
> [4] Hamelijnck, Oliver, et al. "Multi-resolution multi-task Gaussian processes." Advances in Neural Information Processing Systems 32 (2019): 14025-14035.
>
> [5] Domhan, Tobias, Jost Tobias Springenberg, and Frank Hutter. "Speeding up automatic hyperparameter optimization of deep neural networks by extrapolation of learning curves." Twenty-fourth international joint conference on artificial intelligence. 2015.
>
> [6] Klein, Aaron, et al. "Learning curve prediction with Bayesian neural networks.", ICLR 2017.
>
> C6: “The authors state they fit models by ‘looking at the trace plots’, which does not seem particularly robust / automatable…”
>
> R6: First, “looking at the trace plots” is a commonly used trick to access the mixing (burn in) of the MCMC sampler (http://sbfnk.github.io/mfiidd/mcmc_diagnostics.html). Second, we did not look at the trace plots for every training procedure. We used this trick to identify how many steps of HMC are enough, which is 5K in our problems. We then used this number in the following BO experiments.
>
> The leapfrog step-size was tuned manually, from 0.01:0.001:0.02. Note that the step-size over 0.02 does not work well.  If it is below 0.01, the mixing is very slow. Like you have to carefully tune the learning rate of SGD (for stochastic training of NNs), there is no free launch of HMC either.

---

> > ### Author Response · Authors · 2021-08-10
> > **To Reviewer 2(zV6Z) continued**
> >
> > C7: Relation to the asynchronous variant in (Takeno et al. 2020) and GIBBON batch MVES approximation)
> >
> > R7:  (Takeno et al. 2020) proposed two parallel querying strategies. Both strategies leverage the property that the (co-)variance of the conditional Gaussian does not rely on the value of the conditioned variable; so, there is no need to worry about conditioning on function values that are still in query. The asynchronous variant generates new queries conditioned on different sets of function values in query (asynchronously). However, if the conditional parts are significantly overlapping, which might not be uncommon in practice, there is a risk of producing redundant or even collapsed queries. Takeno et al. 2020 also talked about a synchronous version. While they have discussed how to compute the information gain between the function maximum and the function values of a batch of queries, they did not provide a solution of optimizing it with the multi-fidelity query costs. Instead, they suggested a simple heuristics to sequentially find each query by conditioning on the generated ones. However, there is no guarantee about this heuristics.
> >
> > By contrast, our batch multi-fidelity acquisition function explicitly considers the correlation within the query batch, penalizes high correlations, encourages diversity and account for the costs (see Line 156-162).  Hence, our method can effectively avoid the sample collapse or redundance issue. Our fidelity-wise alternating optimization can efficiently maximize the acquisition function and guarantees to improve at every step. The complexity is linear with the number of fidelities, rather than a native optimization that has exponential cost with the number of fidelities.
> >
> > Thanks for providing the reference of GIBBON. This work is concurrent with our work. Note that our paper is a resubmission; our work is accomplished before the ArXiv date of GIBBON. The batch acquisition function in GIBBON coincides with our paper. This confirms the benefit of our proposed acquisition function. However, GIBBON is based on GP surrogate models, and the approximation and optimization of the batch acquisition function is totally different from our work. We will add detailed discussion in our paper.
> >
> > C8: The approach is based on *Laplace approximation, which is presumably a pretty coarse approximation. It's not clear how well the Laplace approximation works for this problem*.
> >
> > R8: **Our approach is *NOT* based on Laplace approximation**. From the model inference, function posterior estimation, acquisition function and optimization, there is not any component related to Laplace approximation.
> >
> > C9: *The difference between BMBO-DARN and DNN-MFBO isn't particularly large* (would be good to see p-values of a hypothesis test for the difference)
> >
> > R9: Actually, as shown in Fig. 1, in all the problems, our method (BMBO-DARN) and DNN-MFBO are significantly different, both from beginning to the late stage (saturation). Note that BMBO-DARN is the red line while DNN-MFBO is the green line. The visual difference is obvious. The black line, BMBO-DARN-1, is close to DNN-MFBO. This is reasonable, because BMBO-DARN-1 indicates the batch size is only one, which essentially performs the same one-by-one querying as DNN-MFBO. The difference between BMBO-DARN and BMBO-DARN-1 also demonstrates the effectiveness of our batch querying method.
> >
> > Second, the results in Fig. 1 are all averaged over five runs. We also provide the error bars. Note that for quite a few methods, their error bars are too small and covered by the markers. The non-overlapping error bars of BMBO-DARN and DNN-MFBO have already indicated the significant statistical difference (p-value < 0.05) [7]
> >
> > [7] Minka, Tom. "Judging significance from error bars." CMU Tech Report (2002).
> >
> > C10: “…lack of detail on the evaluation criteria… not clear to me what exactly the accumulated cost is, and how it is computed in the batch setting”
> >
> > R10: The evaluation criteria have already been given in Line 322-324, 338/346, 351-352, 360, 376-381. These evaluations are all standard. The accumulated cost is the total time of running the ML models, which is easy to count, no matter the batch or single query setting. Note that this is the total querying cost, not including the cost of finding the queries (i.e., training surrogates and optimizing the acquisition function). We compared and reported the query finding cost in Fig. 2.
> >
> > C11: “queries at one input and fidelity each time (B = 1)" performs worse thatn BMBO-DARN is surprising...”
> >
> > R11: We believe this is normal and as expected. Sequentially generating each query without considering the prior ones can easily lead to highly correlated queries or proximal queries in local regions, which contain redundant information, and lower the optimization efficiency. Instead, effective batch queries can reduce the query correlation, enhance the diversity and more easily get over the local regions. Accordingly, the optimization performance/efficiency can be improved. See Line 143-145 for our analysis.
> >
> > C12: Why don't the losses all start at the same value if the initial points are shared between methods? How many function evaluations are actually performed in the experiments?
> >
> > R12: We report the comparison since the first BO step. We did not report the initial points, because every method was the same.
> >
> > C13: “"running L-BFGS with 10, 100, 50K maximum iterations": Is this correct? this seems like an obscene difference in fidelities”
> >
> > R13: We are confused about “obscene difference”. Can you please elaborate on it? The fidelities are referred as to PINN is initially trained, partially trained and fully trained. This is done by restricting the maximum number of iterations in LBFGS, which obviously have different accuracies and costs.
> >
> > C14: Timings in Figure 2: Is this the average time per candidate or the average time per batch?
> > How much time is spent on model fitting, how much on acqf optimization?
> >
> > R14: It shows the average time per candidate, which includes both the “model fitting” and “acaf optimization”.  We will supplement the separate training and optimization time.
> >
> > C15: What MF-MES implementation is used? The paper just states "We used a Python implementation of MF-MES and MF-GP-UCB". It seems from Figure 2 that whatever was used is extremely slow; it seems quite possible to substantially reduce this time (and likely also optimization performance) with a proper implementation.
> >
> > R15: The implementation is consistent with original implementations. Note that for a fair comparison, we used the Python version (consistent with ours). The key bottleneck of MF-MES is that it uses DIRECT (https://github.com/andim/scipydirect) for global optimization, which is known to be very slow.
> >
> > C16: Is there an assumption that the fidelities are ordered? what if that's not the case?
> >
> > R16:  To our knowledge, the existing work all assume the fidelities reflect some kind of ordering, in terms of accuracy, reliability, resolution, etc. Our work is consistent with this common setting. However, it will be interesting to consider unordered fidelities in the future work, especially, what are the corresponding real-world applications, can the existing methods be reusable, etc.

---

### Official Review · Reviewer_JVN7 · 2021-07-20

**Rating:** 4
**Confidence:** 4

**Summary:**

The paper introduces a series of Bayesian neural networks that predict the performance across different, discrete fidelities. The models for higher fidelities use the prediction on lower ones as inputs, which makes this model auto-regressive. the authors use this flexible model and combine it with a new acquisition function that aims to minimize the mutual information of a batch of potential query points and the current belief over the optimum. By designing a heuristic to optimize the acquisition function, the authors are able to apply their model to the problem of Batch BO.

In the experiments, the proposed method is compared to other state-of-the-art methods for regular BO, batched BO and multifidelity BO.

**Ethical Concerns:**

No.

**Limitations And Societal Impact:**

The authors failed to address any limitations of the method. A discussion about the choices of the fidelities, and what happens if they are poorly chosen is missing. A discussion about the overhead of the model would also be appreciated.

**Main Review:**

 I think the paper contains a very interesting method for modeling multiple fidelities/tasks with a cascade of Bayesian neural networks. Based on the presented results. the method seems very sample efficient, but the overhead is unclear. To my surprise, the sequential version of the algorithm performs worse than the batch one, which I would like to hear the authors' feedback on.
Missing related work on batch BO and missing details on the parallelism of the presented experiments, also lower my impression of the paper a bit.
I think that the paper contains two orthogonal ideas: the DARN model itself, and a clever way to approximate the mutual information for multiple points. Because the effects of those are not separated by an ablation study, the effects of contributions both make are not separable.

The paper is easy to follow, although I would suggest to move some of the details of the benchmarks to the appendix and expand the discussion about the acquisition function (which I found too dense) and a more detailed analysis of the results.

## Specific remarks/questions:

- In the discussion of equation (2), I am missing the OG multi-task BO reference

  Swersky, K., Snoek, J., & Adams, R. P. (2013). Multi-task bayesian optimization.

  equation 2 is one particular way of approximating the MTBO kernel, which is more general.
- The discussion about the limitations of multi-fidelity BO is missing MTBO and potentially stacked GPs from:

  Golovin, D., Solnik, B., Moitra, S., Kochanski, G., Karro, J., & Sculley, D. (2017, August). Google vizier: A service for black-box optimization. In Proceedings of the 23rd ACM SIGKDD international conference on knowledge discovery and data mining (pp. 1487-1495)

  which shows as a different way of approximating the full covariance across tasks.
- About equation (5): In the normal context of batch BO, the whole batch is suggested at once, then evaluated and the results incorporated into the model. If this pre-batch query and model-update is followed here, wouldn't $max \lambda_{m_k}$ be more meaningful if the evaluations are in parallel? Otherwise querying a batch wouldn't give any advantage.
- The related work section is missing a discussion about batch BO methods, including (but not limited to):

  Wu, J., & Frazier, P. (2016). The parallel knowledge gradient method for batch Bayesian optimization. Advances in Neural Information Processing Systems, 29, 3126-3134.

  González, J., Dai, Z., Hennig, P., & Lawrence, N. (2016, May). Batch Bayesian optimization via local penalization. In Artificial intelligence and statistics (pp. 648-657). PMLR.

- The method labeld as SHTL, is referred to as ABLR (adaptive Bayesian linear regerssion) in the paper itself. The method also uses L-BFGS to optimize all parameters, which was replaced by ADAM here. Not sure how much that impacts performance, but it should be pointed out in the paper.

- It is nowhere mentioned if evaluations where carried out in parallel. If they were, it should be stated how many parallel evaluations where performed. If they were all done sequentially, I would like to understand why the BMBO-DARN-1 performs worse. Usually, batch methods are outperformed by their sequential counterpart when the batch is evaluated sequentially, but become useful when the evaluations can be performed in parallel.

- BMBO-DARN is very performant across all shown benchmarks, but I wonder how much each contribution has, i.e. how does the auto regressive model compare against a simpler model, but with the same acquisition function, and how would the DARN model work with MF-MES.

- The performance is shown versus the  accumulated cost, so I assume the overhead of the HMC sampling and the elaborate acquisition function is not included. Especially for the XGBoost example, I would assume the overhead to be substantial. What would these plots look like if this overhead is included?

- Following up on the previous point, could you please comment on the average evaluation time for the benchmarks at the different fidelities? It is really hard to judge how many configurations have been evaluated across the different benchmarks.

**Time Spent Reviewing:**

3

---

> ### Author Response · Authors · 2021-08-10
> **To Reviewer 1(JVN7):**
>
> Thanks for your detailed and valuable comments. Here are our responses.
>
> C: comment; R: response
>
> C1: To my surprise, the sequential version of the algorithm performs worse than the batch one, which I would like to hear the authors' feedback on
>
> R1: Sequentially generating each query without considering the previous ones can overlook/underestimate the correlations between consecutive queries, leading to highly correlated or proximal queries in local regions (see analysis in Line 143-150 of our paper). These queries contain redundant information and hence can lower the optimization efficiency.
>
> Instead, our batch acquisition function explicitly accounts for the query correlations inside a batch (see Line 156-162), penalizes strongly correlated queries and encourages diversity (while favoring large information gains). Our alternating optimization method efficiently maximizes this acquisition function to jointly find a set of fidelities and inputs. Therefore, we believe the results are as expected, confirming the effectiveness of our method.
>
> C2: the method seems very sample efficient, but the overhead is unclear
>
> R2: We did compare the overhead, including training and acquisition function optimization, with state-of-the-art methods, as shown in Fig. 2. It can be seen that the overhead of our method (per query) is among the smallest.
>
> C3: “Because the effects of those are not separated by an ablation study, the effects of
> contributions both make are not separable.”
>
> R3: Actually, we have conducted such ablation study to examine “the DARN itself” in Sec 6.1. We evaluated the predictive performance of DARN and the surrogate models of other state-of-the-art BO methods in two synthetic benchmark datasets. As shown in Table 1, DARN consistently outperforms the other surrogate models.
>
> Furthermore, in Sec. 6.3, the comparison between of BMBO-DARN and BMBO-DARN-1 (i.e., batch-size one) actually separately examined the other idea orthogonal to BARN, i.e.,  the “way to approximate the mutual information for multiple points”, because since the surrogate model is the same, i.e., DARN. The results confirm the advantage of that idea --- our method can indeed find batch queries with better quality (significant larger benefit/cost ratio) than sequential queries.
>
> C4: About equation (5) … \max\lambda_m be more meaningful if the evaluations are in parallel? Otherwise querying a batch wouldn't give any advantage.
>
> R4: Even the evaluations can be done in parallel, the cost of each evaluation is still there, and usually cannot be ignored. Take AWS as an example. Suppose we conduct two parallel experiments, each running on the three instances, with the usage of (3hrs, 2hrs, 2hrs) and (3hrs, 1hr, 1hr), respectively. The first experiment will definitely be charged more than the second one. Therefore, using \max \lambda_m will give the incorrect cost, and mislead the multi-fidelity optimization, resulting in a suboptimal benefit-cost ratio.
>
> C5: Missing a few references
>
> R5: Thanks for providing these great references. We will cite and discuss them in the paper.
>
> C6: “The method labeled as SHTL, is referred to as ABLR (adaptive Bayesian linear regerssion) in the paper itself. The method also uses L-BFGS to optimize all parameters, which was replaced by ADAM here. Not sure how much that impacts performance, but it should be pointed out in the paper.”
>
> R6: We have requested the authors of ABLR to share the source code, but the authors said the code is proprietary and cannot be shared. We hence followed the paper details to implement our own version. Our experience shows that using LBFGS often runs into numerical issues, and so we switched to ADAM, which is much more stable. We examined the two optimization methods on simple regression tasks, and they made no difference in performance. We will pointed out this in the paper.
>
> C7: It is nowhere mentioned if evaluations were carried out in parallel… I would like to understand why the BMBO- DARN-1 performs worse. Usually, batch methods are outperformed by their sequential counterpart when the batch is evaluated sequentially…
>
> R7: The evaluations were not carried out in parallel. Note that, we added up the cost of every evaluation. So, it does not matter if these evaluations are in parallel or not; the total cost is the same. We believe the reason of better performance of our batch querying is still in that our batch acquisition function explicitly accounts for the query correlations, penalize highly correlated queries, and encourage diversity; and the sequential counterpart overlooks the query correlations, and is risky to generate redundant queries. See R1 for a more detailed discussion.
>
> C8: …how does the auto regressive model compare against a simpler model, but with the same acquisition function, and how would the DARN model work with MF-MES?
>
> R8: Such comparison is actually between BMBO-DARN-1 and MF-MES, because when batch size is one, our acquisition function is the same as in MF-MES. As we can see from Fig. 1, BMBO-DARN-1 consistently outperforms MF-MES. The improvement is particularly large for XGBoost and PINN (Fig. 1c and d).
>
> In addition, in Sec 6.1, we compared DARN model with MF-MES in terms of prediction accuracy, i.e., surrogate learning performance. As shown in Table 1, DARN model outperforms MF-MES by a large margin.
>
> C9: I assume the overhead of the HMC sampling and the elaborate acquisition function is not included. Especially for the XGBoost example, I would assume the overhead to be substantial. What would these plots look like if this overhead is included?
>
> R9: Yes, the accumulated cost only includes the evaluation/running of the machine learning models. However, we did compare the query generation cost (i.e., overhead), including the HMC sampling and acquisition function optimization, as shown in Fig. 2. As we can see, the overhead of our method is the lowest or very close to the lowest. See R2.  Hence, if we count the overhead in Fig. 1, the plots will not change much, and our method is still advantageous over the competing methods. We will supplement such figures in the paper.
>
> C10: Could you please comment on the average evaluation time for the benchmarks at the different fidelities? It is really hard to judge how many configurations have been evaluated across the different benchmarks.
>
> R10: The average evaluation time at different fidelities is task specific. For example, running the highest fidelity for CNN takes 2 hours while for PINN takes over 1hr on the Nvidia V100 GPU. We will supplement the evaluation time for each benchmark and report the actual number of configurations evaluated by each method at the end of the curves in Fig. 1.

---

> > ### Comment · Reviewer_JVN7 · 2021-08-16
> > **Follow up questions**
> >
> > Thanks for your detailed response and for clarifying most of questions. I still have three things based on your comment that I would still like to comment on
> >
> > 1. In R1 and R7 you point out that all evaluations where done sequentially. You attribute the superior performance of the batched version DARN to the explicit penalization of  the correlation across the samples in a batch vs. the sequential strategy. But shouldn't the updated posterior after each sequential sample in the latter case also take this into consideration? How can you be sure that the improved performance of the batched version is not due to poor model adaptation (e.g., the model is overly confident or the confidence intervals don't shrink over the course of the optimization)? I understand that the Results in Table 1 are supposed to show this, but I find it very hard to judge, how the model predictions change with every additional data point.
> >
> > 2. Based on the average evaluation times you mention in R10, I have to assume that the optimization with BMBO-DARN for the CNN and PINN task does not improve after a time that is not even a full evaluation on the highest fidelity. This could be due to a few different reasons:
> >   - the fidelities on the benchmark are not picked in a way that it takes an evaluation on the highest fidelity to reach the best score
> >   - the shown performance is far from the best that is achievable
> >   - the plots show the performance of the best found configuration at that point in time, but evaluated on the highest fidelity (but that does not seem to be the case, as far as I understand your experimental setup)
> > What are your thoughts on that?
> >
> > 3. Based on the shown overhead of BMBO-DARN in Figure 2., I assume that the overhead for the XGBOOST benchmark is a dominant factor and evaluating a setting is much cheaper than the overhead of your method (even on the highest fidelity). In that case, I would expect the wallclock plot including the overhead to look very different, because Hyperband has almost no overhead and should evaluate several settings while the more elaborate optimizers are still computing the first settings. In this context, I could also consider random search on the highest fidelity to be a very competitive baseline.

---

### Decision · Program_Chairs · 2021-09-27

**Decision:**

Accept (Poster)

**Comment:**

Ultimately, I am recommending this paper for acceptance because I believe the approach to multi-fidelity optimization proposed by the authors is sufficiently novel and the experimental evaluation is sufficiently strong, and I am thus in agreement with Reviewer gu5R and Reviewer Hcby that, after substantial incorporation of feedback from the reviewers, the paper is solid. Furthermore, I believe that you do indeed already include several of the ablation studies asked for (e.g., BMBO-DARN-1 vs MF-MES is equivalent to an ablation of how much the DARN matters). While I believe the paper has flaws (see below), I believe most of these flaws are in argumentation rather than in execution, and would not require changes significant enough to warrant rejection.

With that said, I want to be clear that several of the criticisms raised by Reviewer zV6Z and Reviewer JVN7 seem valid to me, and for at least a few of them, I found the author response to be unconvincing.

First, the claims made on lines 143-149 and repeated in the author feedback that it is expected for batch acquisition to be more query efficient than sequential acquisition seem specious to me, and I would remove or rephrase them. If I am selecting a set of points x_1,...,x_k, clearly there is nothing inherent about knowing the labels y_1,...,y_{k-1} that forces a worse choice of x_{k}. Rather, certain sequential acquisition functions like EI are known for being myopic. Having access to more information should always enable more informed decisions than having access to less information. Saying that sequential decision making can "underrate the correlations between consecutive queries" does not seem rigorous enough to explain this claim. Rather, batch acquisition is typically more efficient than sequential optimization in terms of total wall-clock time, and indeed the results presented in Figure 1 are given in terms of time. While the advantages of your approach in terms of time elapsed are clear, I would **strongly recommend** the authors additionally use the supplementary materials to report results in terms of query efficiency to remove this confusion.

Second, your claim that "our topic/work is multi-fidelity batch BO and is irrelevant to early-stopping" seems overly dismissive of Reviewer zV6Z's concern. Every experiment in Figure 4 falls precisely into the setting Reviewer zV6Z is describing, where training can be stopped at any time (e.g., early stopping) rather than pre-determining a set of training lengths to use as epochs, as for example Freeze-thaw BO (Swersky et al., 2014) does, which you yourself cite. While of course in the general setting, fidelities can be arbitrary discrete measures of precision that do not lend themself to early stopping, the experiments you choose to run in your paper do indeed fit the setting considered by Swersky et al. 2014 and others. I would strongly encourage the authors to reconsider which criticisms may be valid and which may be dismissed when preparing the camera ready version, as many of the questions raised here are likely to be shared by readers of the paper at large.